# Head movements affect skill acquisition for ball trapping in blind football

Takumi Mieda [1]*, Masahiro Kokubu [2]

1 Faculty of Education, Art and Science, Yamagata University, Yamagata, Japan, 2 Institute of Health and Sport Sciences, University of Tsukuba, Tsukuba, Japan

* mieda@e.yamagata-u.ac.jp

## Abstract

Blind football players use head movements to accurately identify sound location when trapping a ball. Accurate sound localization is likely important for motor learning of ball trapping in blind football. However, whether head movements affect the acquisition of ball-trapping skills remains unclear. Therefore, this study examined the effect of head movements on skill acquisition during ball trapping. Overall, 20 sighted male college students were recruited and assigned to one of the following two groups: the conventional training group, where they were instructed to move leftward and rightward to align their body with the ball's trajectory, and the head-movement-focused group, where they were instructed to follow the ball with their faces until the ball touched their feet, in addition to the conventional training instructions. Both groups underwent a 2-day training for ball trapping according to the specific instructions. The head-movement-focused group showed a decrease in errors in ball trapping at near distances and with larger downward head rotations in the sagittal plane compared to the conventional training group, indicating that during the skill acquisition training for ball trapping, the sound source can be localized more accurately using larger head rotations toward the ball. These results may help beginner-level players acquire better precision in their movements while playing blind football.

## Introduction

Blind football is a sport designed for the visually impaired. In blind football, localizing sound sources based on auditory information, such as teammates' voices and the sound of the ball rolling, is crucial for determining the distance and direction of the source. Accurate sound localization is essential for the success of auditory-motor tasks during training and games. In particular, the success of trapping an approaching ball depends on the ability to judge its ever-changing position, rather than dribbling and other motor skills that require a technical element to handle the ball.

Playing blind football long-term has been shown to improve sound localization and auditory-motor skills that require sound localization. For example, blind football players are more accurate and faster at localizing the sound using a task to localize sound sources presented by multiple static loudspeakers [1–3]. Recently, blind football players have been shown to be

**Funding:** JSPS KAKENHI (Grant Numbers: 22K17728, 19J13848). Cooperative Research Grant of Advanced Research Initiative for Human High Performance, University of Tsukuba. The funders had no role in study design, data collection and analysis, decision to publish, or preparation of the manuscript.

**Competing interests:** The authors have declared that no competing interests exist.

more accurate in trapping an approaching ball (i.e., a moving sound source) than sighted individuals inexperienced in playing blind football, suggesting that blind football players can localize moving sound sources more accurately [4]. However, no evidence exists regarding the effects of head movements on ball-trapping skill acquisition in blind football.

A previous study [5] showed improvements in sound localization during spatial training. This indicated a role of both the head movements used to rotate (i.e., orient) their heads towards the direction of the sound source and head movements that are not necessarily directed towards the sound but serve to inspect the acoustic space and increase the possibility of extracting auditory cues, such as dynamic cues [6–8]. These findings suggest that such head movements were used strategically to localize sound sources during monaural listening training.

In fact, head movements have been shown to enhance the accuracy of sound localization [6–11]. Additionally, previous studies have used the task of localizing static loudspeakers to investigate the effect of head movements on source localization accuracy. For instance, compared to the fixed-head condition, the head-rotatable condition reportedly increases sound localization accuracy in the horizontal [7, 8] and vertical [6, 10] planes. However, these previous studies mainly used tasks involving the localization of fixed sound sources. More recently, the impact of head movements on sound source localization has been investigated in tasks aimed at localizing moving sound sources. For instance, the minimum audible angle threshold has been observed to be smaller when listeners move their heads while the presented sound stimulus is in motion than when they remain stationary [12]. Based on these previous studies' findings, head movements appear important for accurately localizing moving sound sources during ball trapping in blind football.

Orientating the head to the sound source is necessary to accurately localize the source [13] and facilitate the actions associated with it. Ball trapping in blind football is a combination of locating the spatial position of the ball according to the player's body and subsequently moving the legs toward the ball. In general, the spatial position of an object is coded with frames of reference, which are either egocentric, with the body as the reference point, or allocentric, with an arbitrarily determined location in space as the reference point [14]. Regarding the relationship between perception and action, because the egocentric frame of reference provides the coordinate system necessary for performing goal-oriented actions such as reaching and grasping [15, 16], it can be inferred that source localization with the egocentric coordinate system is significant in ball trapping. Furthermore, rotating the head towards the direction of the sound presented by static loudspeakers provides an accurate reference frame to control pointing to the sound source [17]. Accordingly, in auditory-motor tasks, such as ball trapping, head rotation towards the ball provides an accurate frame of reference and improves skills. However, the effects of head rotation on auditory-motor skills, such as trapping a moving sound source, have not been fully examined.

Reviewing previous studies, head–ball coupling is reportedly a specific strategy observed in skilled players during visuomotor tasks in which they couple the head direction with the ball's movement to track the target [18–21]. According to a previous study, elite baseball batters rotate their heads until bat-to-ball contact, indicating that they may rotate their heads to predict ball trajectories [19]. Similarly, elite cricket batters have a tight head–ball coupling, indicating an advantage in keeping the target in a consistent direction relative to the head since visuomotor tasks, including catching and hitting, are controlled egocentrically [21]. Given that the location of the sound source is mainly referenced in the head-centered frame of reference [22, 23], it is crucial for skill acquisition in auditory-motor tasks, such as ball trapping, to track the ball in a consistent direction relative to the head. A previous study has demonstrated that blind footballers couple downward head rotation with the movement of an approaching ball,

ensuring the ball remains in a consistent egocentric direction relative to the head from an early point after ball launching to the moment of ball trapping [4]. However, it is unclear whether ball trapping performance would be improved with training that instructs to follow the ball with their heads facing in the ball's direction throughout ball trapping.

Therefore, the present study aimed to address the gap in the literature regarding possible improvement in auditory-motor skills through head movements by clarifying the effect of such movements on the acquisition of ball-trapping skills in blind football. Particularly, we examined how ball-trapping performance improved before and after the acquisition of skills by comparing a conventionally trained group, instructed to move leftward and rightward to align their body with the trajectory of the ball, with a head-movement-focused group, instructed to follow the ball with their faces until the ball touched their feet, in addition to conventional training instructions. Blind football players who employ a strategy of localizing the sound source by turning their heads towards the ball have demonstrated superior performance in ball trapping, as evidenced by small absolute and variable errors (AEs and VEs, respectively) [4]. In the present study, we expected that AEs and VEs would be reduced in the head-movement-focused group compared to those in the conventionally trained group.

A previous study [24] investigated reaching accuracy to different sound source positions in the peri-personal space, showing better reaching accuracy in azimuth toward the targets for near distances (0.72–0.80 m) compared to that for far distances (0.88–1.08 m). Given that the accuracy of reaching the sound source varies with distance, it is reasonable to expect that the effect of training with head rotations towards the ball in ball trapping, as an auditory-motor task, would similarly differ depending on the distance. Here, we investigated whether the training effect would vary with different distance conditions.

We hypothesized that ball-trapping performance with respect to AEs and VEs would improve in the head-movement-focused group compared to that in the conventional training group, especially in the near-distance condition. We also hypothesized that, compared to the conventionally trained group, the head-movement-focused group would couple downward head rotation with the movement of an approaching ball throughout ball trapping.

## Materials and methods

### Participants

The recruitment period for this study was from 2 October 2020 to 13 July 2021. The participants enrolled were sighted male college students (n = 20) with a mean age of 23.0 ± 1.9 years without regular blind football training or football training. As females have been reported to perceive sounds more closely than males [25, 26], only males were sampled in this study to eliminate the influence of the sex difference. The participants were divided into two groups: a conventional training group (n = 10) and a head-movement-focused group (n = 10). Stratified randomization was used to ensure an equal distribution of age. Sample size was driven by previous studies that investigated the effect of audio-motor training on sound localization [27] (n = 7) and body motion capture [28] (n = 10) especially during passing/receiving the ball in blind football [29] (n = 10) for sighted individuals. The two groups were given different instructions for skill acquisition in ball trapping (*see sections titled Task and Procedure*). The preferred leg of participants in both groups for kicking a ball was noted, and a history of possible disorders was self-declared based on a previous study (1). Furthermore, we confirmed that all participants were right-footed and had no history of hearing deficiencies. This study adhered to the latest version of the Declaration of Helsinki and was approved by the Ethics Committee of the Faculty of Education, Art, and Science, Yamagata University, and the

Institute of Health and Sport Sciences, University of Tsukuba. Written informed consent was obtained from all participants before their involvement in this study.

## Apparatus

The apparatus and tasks used in this study are based on a previous study (4). An overhead view of the experimental setup is shown in Fig 1. A non-anechoic room (7.3 m × 7.3 m × 2.5 m) was used for the experiment under a reverberant listening condition [3, 29]. This approach has an ecological focus as it accurately reflects the conditions of playing blind football. The joint mats (B0013HF3DS; TOEI LIGHT Co., Ltd., Tokyo, Japan) were placed on the floor of the room to prevent fall-induced injuries. Thin threads were attached to two locations on the surface of the mat so that the participants could confirm the original position. An intermediate position between the two locations was defined as the reference point. An infrared motion capture system including 11 cameras (OptiTrack Flex13, 120 fps; Natural Point, Inc. Corvallis, OR, USA) was used to measure the three-dimensional position of the reflective markers attached to the body parts of each participant. In order to obtain an optimal record of the spatial position of the participant's body and the ball, at least three cameras were placed to allow simultaneous focus on every infrared reflective marker (see Fig 1). Next, 16 infrared reflective

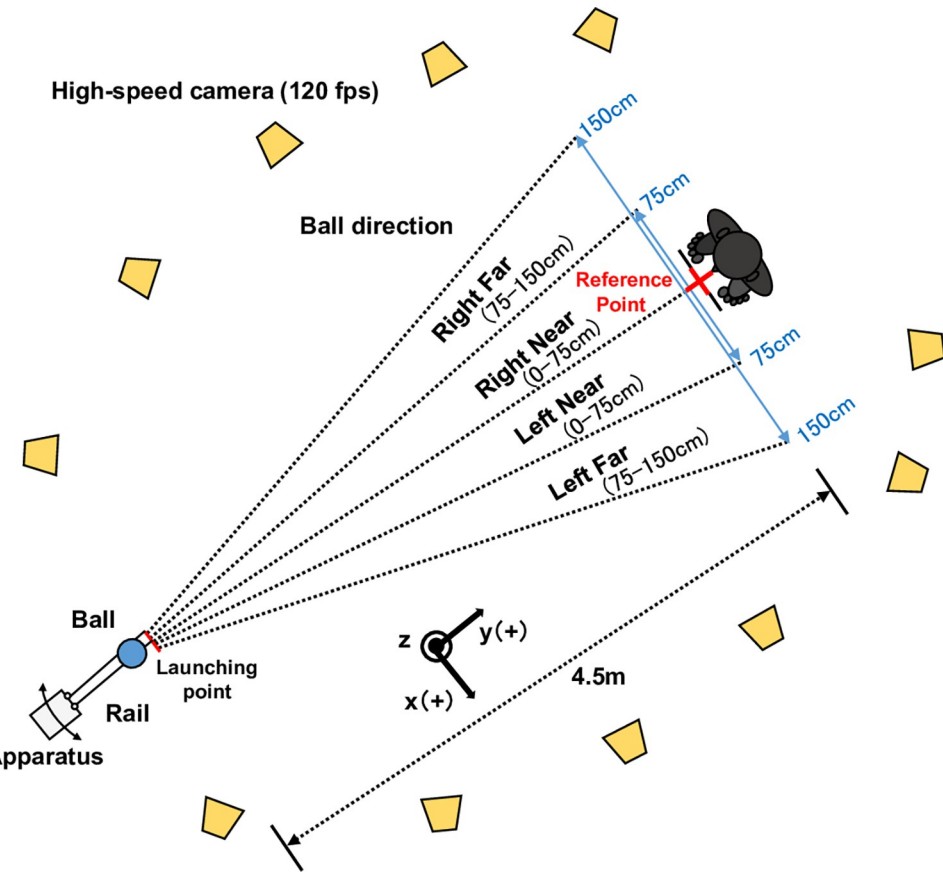

**Fig 1. Experimental set-up.** Reflective markers were attached to the body of each participant. The three-dimensional position of the reflective markers was computed using an infrared camera motion system with 11 cameras. The cameras were placed around the ball and the participant's body. The ball was launched in four different directions (right far, right near, left far, and left near). The near and far spaces ranged 0–75 and 75–150 cm, respectively, laterally away from the reference point. The figure was adapted and revised from a previous study [4].

markers, 14 mm in diameter, were attached to each participant's body. Specifically, three infra-red reflective markers were attached to the parietal region and each tragion, with the parietal marker serving as a triangular tip [4, 30]. Other reflective markers were attached as follows: one marker on the superior margin of the sternum (breastbone), one marker on each toe, heel, lateral malleolus, lateral epicondyle of the femur, greater trochanter, shoulder, and one on the breastbones.

This study used an official ball approved by the Japan Blind Football Association. The ball was attached with an infrared reflective tape to measure the ball's position. The experimenter launched the ball at the participants by using a custom-built apparatus including an aluminium rail that allowed for ball speed regulation and direction. The rail attached to the apparatus was set at an inclination of 45˚ with respect to the floor. The distance between the launching point of the ball and the reference point was 4.5 m. We set three different height positions on the rail to obtain ball speeds of 2.6, 2.3, and 1.9 m/s, and released the ball manually from one of the positions. In the previous study [4], speed was not treated as a dependent variable, but different speed conditions were established to prevent the learning effect of the participants. Consistent with this approach, the present study also randomized the speed to minimize the learning effect, with the aim of reducing its potential impact on AEs and VEs in ball trapping. We adjusted and altered the direction of the rail for every trial so that the participants could not anticipate the direction of the ball before launching. Additionally, the rail was moved on each trial by the experimenter to intentionally generate noise, even when the ball was continuously in the same direction, thereby preventing participants from knowing the direction of the ball before launching. Four directional conditions (right far, right near, left far, and left near) were set as shown in Fig 1. The near and the far distances ranged 0–75 and 75–150 cm, respectively, laterally away from the reference point. The combinations of ball speed and trajectory were administered in a randomized order, following a predefined order for all participants. Furthermore, in all trials, the same experimenter moved the device to ensure consistency and to control for any variability.

## Task

The participants were instructed to cover their eyes with an eye mask when performing the task and to stand in their original positions. They were also instructed to trap the approaching ball using their right feet as accurately as possible. Once the participants performed trapping, they returned to their original positions for the next trial. Next, they were instructed to move leftward and rightward to align their bodies along the ball's trajectory during the experiment since this is a fundamental movement for ball trapping during the first stage of beginners training [31]. In the acquisition and post-test, the head-movement-focused group was instructed to follow the ball with their faces until it touched their feet. This instruction was included in this study because turning the face toward a sound is believed to contribute to better sound localization [32].

## Procedure

Skill acquisition training for ball trapping was conducted over a total of 2 days: the first day consisted of practice, pre-test, and acquisition, and the second day consisted of acquisition and post-test. A 15-min interval was observed between the pre-test and the skill acquisition phase on day 1, during which approximately 8 min were spent explaining the instructions. On day 2, there was a 7-min interval between the skill acquisition phase and the post-test. The longer interval on day 1, compared to day 2, was implemented to provide participants a break and to ensure that all participants, who were beginners in blind football, had sufficient time to receive and fully comprehend the instructions provided to each group regarding the execution of ball

trapping. The 2-day experiment was conducted on consecutive days for all participants. The participants were asked to perform the ball-trapping task on day 1 during various trials that involved three different speeds (2.6, 2.3, and 1.9 m/s) and four different directions (right far, right near, left far, and left near). For instance, there were 12 trials, with 4 trials for each of the three speeds during the pre-test. They performed 12 practice trials to become familiar with the task, during which the combinations of ball speed and trajectory were administered in a predefined randomized order. Following the practice trials, they completed 12 trials during the pre-test and 48 trials (12 trials × 4 blocks) during the acquisition on day 1. Each block consisted of three different speeds and four different directions, resulting in a total of 12 trials per block. The 4 blocks were all administered in a randomly determined order. Subsequently, they completed 48 trials (12 trials × 4 blocks) during the acquisition and 12 trials during the post-test on day 2. The approximate duration of the skill acquisition training on both day 1 and day 2 was 36 min. No notable differences were observed in the duration between the two groups or among individual participants within each group. The participants rested for 3–5 min as required between the blocks.

## Statistical analysis

The positional data were low-pass filtered with a cutoff frequency of 5 Hz. We computed the kinematic and performance variables using a custom-written MATLAB code (R2023b, Math-Works Inc., Natick, MA, USA). The following kinematic and performance variables were calculated based on the analysis of a previous study [4]; please refer to the cited study for details. Kinematic variables were calculated as the head rotation angle in the sagittal plane ($\theta_{\mathrm{HDs}}$), head rotation angle in the horizontal plane ($\theta_{\mathrm{HDh}}$), and the trunk rotation angle in the sagittal plane ($\theta_{\mathrm{TRs}}$). Moreover, the head–trunk angle in the sagittal plane ($\theta_{\mathrm{HDs}}-\theta_{\mathrm{TRs}}$) was calculated by subtracting the trunk rotation angle from the head rotation angle in the sagittal plane to comprehensively examine whether head rotation was caused by the head or trunk angle. The head rotation angle in the sagittal plane for each trial was normalized to 101 time points (1–101); this was accomplished by resampling the time of each movement using MATLAB's interp1 function. To verify whether the head-ball coupling strategy was used in the head-movement-focused training group, we also calculated the head rotation angles in the sagittal plane relative to those at the time of ball launching for each 10% time point [4]. AEs and VEs for the performance variables were computed based on the distances between the center of the ball and the midpoint between the toe and heel on the right foot at the ball trapping point [4]. AE was determined as the absolute value of the distance on the x-axis between the center of the ball and the midpoint between the toe and heel on the right foot, whereas VE was determined as the standard deviation of the constant error, computed by the distance on the x-axis between the center of the ball and the midpoint between the toe and heel on the right foot.

Regarding the performance and kinematic variables, the mean of the 12 trials in the pre- and post-tests was calculated for each participant. The difference between pre- and post-tests was calculated by subtracting the mean of the post-test from that of the pre-test. For AE and VE, a negative value for the difference indicated that the error decreased from the pre- to post-test. Similarly, regarding the head and trunk rotation angles in the sagittal plane, a negative value for the difference indicated that the head and trunk were rotated more downward from the pre- to post-test. Regarding the head–trunk rotation angle in the sagittal plane, a negative value for the difference indicated that the head was rotated more downward in relation to the trunk from the pre- to post-test. Regarding the head rotation angle in the horizontal plane, a positive value for the difference indicated that the head was rotated more rightward from the pre- to post-test. The Shapiro–Wilk test was performed to determine whether the data were

normally distributed ($p > 0.05$). Bonferroni correction for multiple comparisons was used when appropriate (relative angles of head rotations in the sagittal plane at 10 time points, $p < 0.05/10 = 0.005$). A nonparametric test (two-tailed Mann–Whitney U-test) was performed when the normality assumption was not met. Data from three trials in the conventional training group and six trials in the head-movement-focused group were discarded from the analysis because they were incorrectly recorded. All analyses were conducted using IBM SPSS Statistics for Windows (version 28.0; IBM Corp., Armonk, N.Y., USA). A $p < 0.05$ was considered statistically significant.

## Results

### Performance

Table 1 presents the baseline performance in the pre-test. Regarding AE in the near distance, there was no significant difference (U = 27, Z = -1.74, $p = 0.08$) between the head-movement-focused and conventional training groups. For VE in the near distance, the VE in the head-movement-focused group was significantly higher than that in the conventional training group (U = 22, Z = -2.12, $p = 0.03$). Regarding AE in the far distance, there was no significant difference (U = 45, Z = -0.38, $p = 0.71$) between the head-movement-focused and conventional training groups. For VE in the far distance, (U = 40, Z = -0.76, $p = 0.45$), there was no significant difference between the head-movement-focused and conventional training groups.

Table 1 also presents the differences in performance during the acquisition phase. Fig 2 shows the constant error for one participant in each group in the near distance in the pre- and post-tests.

Regarding the near distance, the reduction in AE in the head-movement-focused group (-8.19 [-13.81 to -3.22 cm]) was significantly greater than that in the conventional training group (-0.08 [-6.28 to 1.86 cm]) throughout the skill acquisition phase (U = 24, Z = -1.97, $p < 0.05$), as shown in Fig 3A. However, there was no significant change in the difference in VE (U = 27, Z = -1.74, $p = 0.08$) between the head-movement-focused group (-6.30 [-10.21 to -5.01 cm]) and the conventional training group (-0.61 [-6.98 to 1.90 cm]) as shown in Fig 3B. Regarding the far distance, there was no significant change in the difference in AE (U = 38, Z = -0.91, $p = 0.36$) between the head-movement-focused group (-1.33 [-5.34 to 1.47 cm]) and the conventional training group (-3.21 [-7.33 to -0.21 cm]) as shown in Fig 3C. Moreover, there was no significant change in the difference in VE (U = 42, Z = -0.60, $p = 0.55$) between the head-movement-focused group (-2.71 [-12.33 to -1.03 cm]) and the conventional training group (-0.48 [-4.82 to 0.82 cm]) as shown in Fig 3D.

**Table 1. Change in performance.**

| | | | Head | | | Conventional | | |
|---|---|---|---|---|---|---|---|---|
| | | | Pre-Test | Post-Test | Amount of Difference (cm) | Pre-Test | Post-Test | Amount of Difference (cm) |
| AE | Near | *Median (IQR)* | 18.78 (17.09 to 27.54) | 13.49 (10.03 to 17.16) | -8.19 (-13.81 to -3.22) | 13.66 (12.12 to 16.16) | 13.36 (9.27 to 14.01) | -0.08 (-6.28 to 1.86) |
| VE | Near | *Median (IQR)* | 24.99 (19.91 to 26.15) | 14.87 (12.86 to 17.39) | -6.30 (-10.21 to -5.01) | 14.89 (10.84 to 18.06) | 12.89 (9.78 to 16.30) | -0.61 (-6.98 to 1.90) |
| AE | Far | *Median (IQR)* | 15.15 (11.55 to 19.35) | 12.33 (10.93 to 14.31) | -1.33 (-5.34 to 1.47) | 12.94 (11.12 to 19.81) | 10.49 (7.43 to 12.61) | -3.21 (-7.33 to -0.21) |
| VE | Far | *Median (IQR)* | 19.48 (13.40 to 24.50) | 13.22 (11.99 to 16.34) | -2.71 (-12.33 to -1.03) | 15.71 (13.82 to 18.38) | 13.42 (9.71 to 15.01) | -0.48 (-4.82 to 0.82) |

The median of the absolute error (AE) and variable error (VE) in the pre-test, post-test, and amount of difference between the head-movement-focused group (Head) and the conventional training group (Conventional).

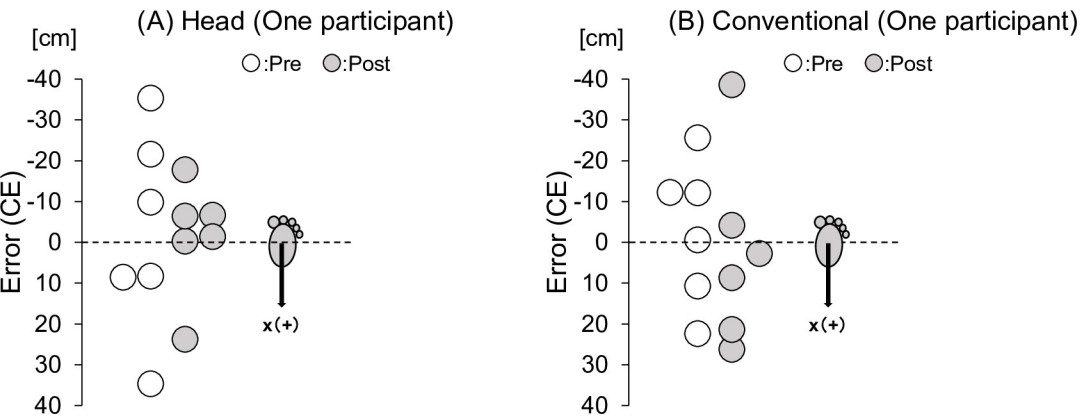

**Fig 2. Typical example error of one participant in each group.** Demonstration of constant error (CE) of one participant in (A) the head-movement-focused group (Head) and (B) the conventional training group (Conventional) in the near distance. The gray and white colored circles represent the CE of each participant in the pre-test and post-test, respectively. The z-axis represents the error for the left and right directions.

## Head rotation angles in the sagittal and the horizontal planes

Table 2 shows the differences in the peak-to-peak amplitudes of head angles in the sagittal and horizontal planes. Regarding the head angle in the sagittal plane, the head-movement-focused group directed their head more downward than did the conventional training group in the

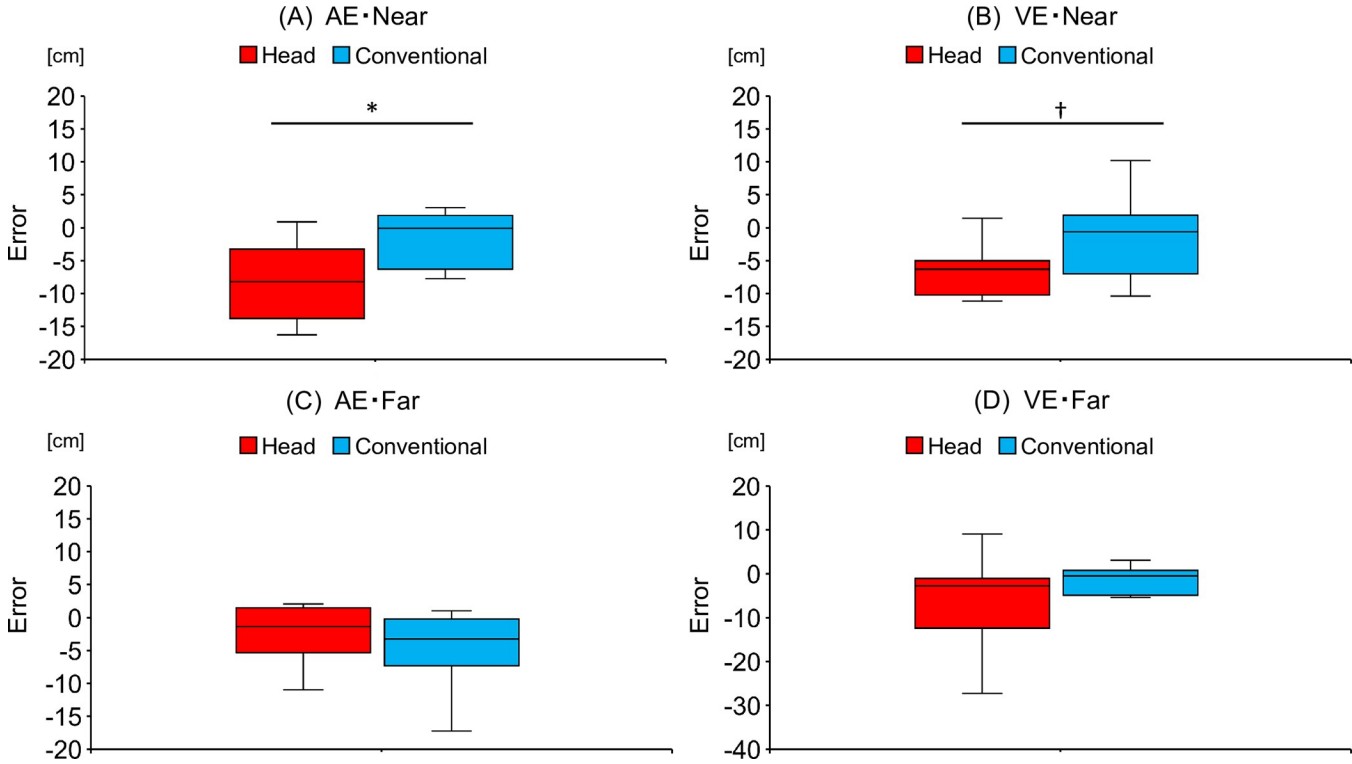

**Fig 3. The amount of difference in absolute error (AE) and variable error (VE) between pre- and post-tests.** Comparison of the amount of difference in (A) AE and (B) VE in the near distance and the amount of difference in (C) AE and (D) VE in the far distance between the head-movement-focused group (Head) and the conventional training group (Conventional). A negative value on the vertical axis in the figure indicates that the error decreased from the pre- to the post-test, while a positive value indicates that the error increased. $^*p < 0.05$.

**Table 2. Change in head angle in the sagittal ($\theta_{HDs}$) and horizontal ($\theta_{HDh}$) planes.**

| | | | Head | | | Conventional | | |
|---|---|---|---|---|---|---|---|---|
| | | | Pre-Test | Post-Test | Amount of Difference (deg.) | Pre-Test | Post-Test | Amount of Difference (deg.) |
| Head ($\theta$HDs) peak-to-peak amplitude | Near | *Median (IQR)* | -13.54 (-21.32 to -8.13) | -35.03 (-38.70 to -30.39) | -21.50 (-25.58 to -11.63) | -15.24 (-20.95 to -10.87) | -18.81 (-20.41 to -16.99) | -2.94 (-6.98 to -0.07) |
| Head ($\theta$HDs) peak-to-peak amplitude | Far | *Median (IQR)* | -12.08 (-20.82 to -8.26) | -31.83 (-35.53 to -29.19) | -20.51 (-22.15 to -8.82) | -16.15 (-25.95 to -13.11) | -18.83 (-20.25 to -15.19) | -0.79 (-4.63 to 1.43) |
| Head ($\theta$HDh) peak-to-peak amplitude | Near | *Median (IQR)* | 18.04 (12.51 to 26.92) | 24.21 (15.53 to 29.85) | 6.22 (-2.11 to 10.86) | 12.65 (10.66 to 19.23) | 19.02 (12.38 to 21.81) | 2.53 (1.17 to 6.98) |
| Head ($\theta$HDh) peak-to-peak amplitude | Far | *Median (IQR)* | 22.55 (19.79 to 29.13) | 27.18 (22.07 to 36.06) | 3.67 (0.77 to 8.06) | 19.46 (13.37 to 22.35) | 20.87 (16.71 to 24.38) | 1.42 (-0.94 to 4.40) |

The median of the head angles in the sagittal and horizontal planes in the pre-test, post-test, and the amount of difference between the head-movement-focused group (Head) and the conventional training group (Conventional).

near (U = 6, Z = -3.33, p < 0.001) and far (U = 2, Z = -0.38, p < 0.001) distances during the skill acquisition phase, as shown in Fig 4A and 4B. The results were as follows: in the near distance, head-movement-focused group (-21.50 [-25.58 to -11.63 deg.]) vs. conventional training group (-2.94 [-6.98 to -0.07 deg.]); in the far distance, head-movement-focused group (-20.51

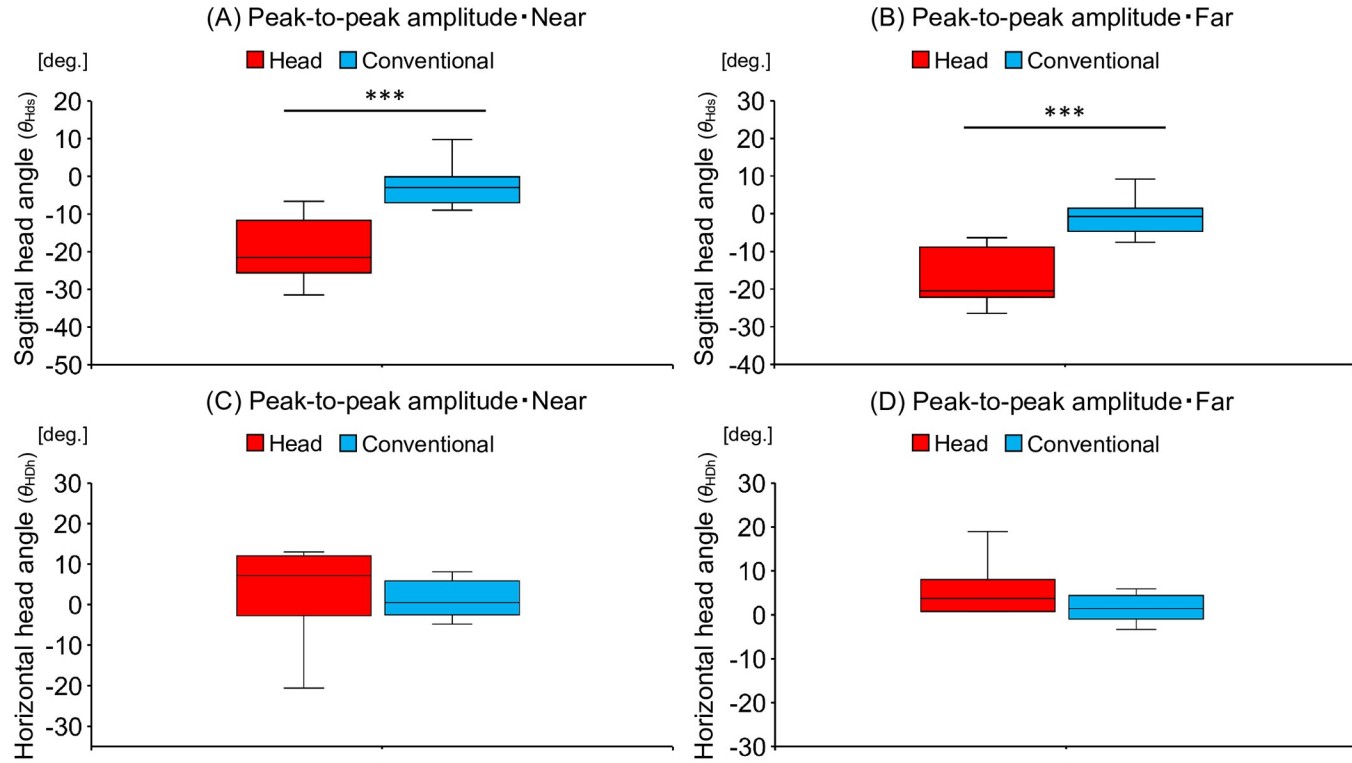

**Fig 4. The amount of difference in head rotation angles in the sagittal ($\theta_{HDs}$) and horizontal ($\theta_{HDh}$) planes between pre- and post-tests.** Comparison of the amount of difference in peak-to-peak amplitude of head rotation angle in the sagittal plane (A) in the near distance and (B) far distance between the head-movement-focused group (Head) and the conventional training group (Conventional). A negative value on the vertical axis in the figure indicates that the head was rotated more downward from the pre- to the post-test, while a positive value indicates that the head was rotated more upward. Comparison of the amount of difference in peak-to-peak amplitude of head rotation angle in the horizontal plane (C) in the near distance and (D) far distance between the head-movement-focused group (Head) and the conventional training group (Conventional). A negative value on the vertical axis in the figure indicates that the head was rotated more leftward from the pre- to the post-test, while a positive value indicates that the head was rotated more rightward.***p < 0.001.

[-22.15 to -8.82 deg.]) vs. conventional training group (-0.79 [-4.63 to 1.43 deg.]). Head angles in the horizontal plane in the near (U = 45, Z = -0.38, p = 0.71) and far (U = 41, Z = -0.68, p = 0.50) distances did not differ between the two groups, as shown in Fig 4C and 4D. The results were as follows: in the near distance, head-movement-focused group (6.22 [-2.11 to 10.86 deg.]) vs. conventional training group (2.53 [1.17 to 6.98 deg.]); in the far distance, head-movement-focused group (3.67 [0.77 to 8.06 deg.]) vs. conventional training group (1.42 [-0.94 to 4.40 deg.]).

Table 3 shows the mean relative angles of head rotation in the sagittal plane for each group at each time point. The relative angles of head rotation in the sagittal plane at each time point were significantly larger in the head-movement-focused group than in the conventional training group at time points 20%, 50%, 60%, 70%, 80%, 90%, and 100% in the near distance (Fig 5B) and at time points 30%, 40%, 50%, 60%, 70%, 80%, 90%, and 100% in the far distance (Fig 5D) in the post-test phase. However, there was no significant difference in the mean relative angles of head rotation in the sagittal plane at any time point between the head-movement-focused group and the conventional training group in the near distance (Fig 5A) or far distance (Fig 5C) in the pre-test phase. These results indicate that downward head rotations were consistently observed throughout the ball-trapping phase in the head-movement-focused group compared to the conventional training group in the post-test rather than in the pre-test phase.

## Trunk rotation and head–trunk angles in the sagittal plane

Table 4 shows the differences in the peak-to-peak amplitudes of the trunk and head–trunk angles in the sagittal plane. Regarding the trunk angle in the near distance, the head-movement-focused group (-3.45 [-4.65 to -1.79 deg.]) tilted their trunk downward more than did the conventional training group (-0.23 [-1.30 to 0.70 deg.]) during the skill acquisition phase (U = 18, Z = -2.42, $p < 0.05$) as shown in Fig 6A. Similarly, for the far distance, the head-movement-focused group (-4.02 [-7.14 to -1.34 deg.]) tilted their trunk downward more than did the conventional training group (0.92 [-0.76 to 2.15 deg.]) during the skill acquisition phase (U = 14, Z = -2.72, $p < 0.01$) as shown in Fig 6B.

Regarding the head–trunk angle, in the near distance, the head-movement-focused group (-16.07 [-21.54 to -14.24 deg.]) directed their head more downward relative to the trunk, when compared to the conventional training group (-1.44 [-7.24 to -0.57 deg.]) during the skill acquisition phase (U = 12, Z = -2.87, $p < 0.01$) as shown in Fig 6C. Similarly, in the far distance, the head-movement-focused group (-13.82 [-18.69 to -9.63 deg.]) directed their head more downward relative to the trunk, when compared to the conventional training group (-3.04 [-4.54 to -0.98 deg.]) during the skill acquisition phase (U = 16, Z = -2.57, $p < 0.05$) as shown in Fig 6D.

## Discussion

The present study aimed to examine the effect of head rotation towards the sound source on skill acquisition of ball trapping in blind football. Particularly, we examined how ball-trapping performance could be improved by comparing the conventional training group instructed to move leftward and rightward to align their body with the ball's trajectory with the head-movement-focused group instructed to follow the ball with their faces until the ball touched their feet, in addition to the conventional training instructions. Regarding performance in the near distance, the reduction in error in the head-movement-focused group was significantly greater than that in the conventional training group (Fig 3). The localization of a sound source according to the listener's head affects the accuracy of performance in auditory-motor tasks, such as

**Table 3. Relative angles of head rotation in the sagittal plane ($\theta_{HDs}$) at each time point.**

|  | Relative Time (%) | Head<br>Median (IQR) | Conventional<br>Median (IQR) | Z | p |  |
|---|---|---|---|---|---|---|
| Near Pre-Test | 10 | -0.22 (-0.30 to 0.00) | -0.29 (-0.48 to -0.21) | -0.91 | 0.364 | |
| | 20 | -0.47 (-1.13 to 0.11) | -0.54 (-1.08 to -0.43) | -0.45 | 0.650 | |
| | 30 | -0.80 (-1.87 to -0.42) | -1.09 (-2.44 to -0.77) | -0.45 | 0.650 | |
| | 40 | -1.44 (-3.07 to -0.88) | -2.23 (-3.68 to -1.23) | -0.53 | 0.597 | |
| | 50 | -3.11 (-5.37 to -1.32) | -4.26 (-6.06 to -1.78) | -0.45 | 0.650 | |
| | 60 | -3.76 (-7.70 to -2.20) | -6.07 (-9.74 to -2.84) | -0.91 | 0.364 | |
| | 70 | -6.25 (-10.94 to -2.99) | -8.46 (-12.47 to -4.39) | -0.76 | 0.450 | |
| | 80 | -9.64 (-15.38 to -3.74) | -11.43 (-15.59 to -5.77) | -0.83 | 0.406 | |
| | 90 | -10.81 (-17.08 to -5.42) | -13.44 (-18.91 to -7.16) | -0.83 | 0.406 | |
| | 100 | -11.78 (-16.46 to -7.14) | -14.49 (-20.46 to -7.86) | -0.53 | 0.597 | |
| Near Post-Test | 10 | -2.21 (-3.53 to -0.30) | -0.26 (-0.44 to -0.08) | -2.04 | 0.041 | |
| | 20 | -4.65 (-6.29 to -1.73) | -0.72 (-1.37 to 0.01) | -2.87 | 0.004 | * |
| | 30 | -6.72 (-9.20 to -3.81) | -2.19 (-2.78 to -0.57) | -2.80 | 0.005 | |
| | 40 | -8.33 (-10.67 to -6.93) | -4.08 (-4.57 to -1.10) | -2.72 | 0.007 | |
| | 50 | -10.66 (-13.04 to -8.79) | -5.81 (-6.88 to -2.18) | -3.10 | 0.002 | * |
| | 60 | -14.25 (-16.34 to -12.66) | -7.06 (-8.69 to -4.45) | -3.40 | 0.001 | * |
| | 70 | -19.20 (-20.55 to -17.53) | -9.19 (-11.44 to -8.62) | -3.48 | 0.001 | * |
| | 80 | -27.02 (-28.11 to -24.51) | -12.80 (-14.82 to -11.56) | -3.17 | 0.001 | * |
| | 90 | -32.16 (-34.90 to -28.98) | -15.95 (-17.98 to -13.56) | -3.17 | 0.001 | * |
| | 100 | -32.85 (-37.81 to -29.75) | -16.70 (-19.92 to -13.39) | -3.10 | 0.002 | * |
| Far Pre-Test | 10 | -0.19 (-0.25 to 0.02) | -0.38 (-0.54 to -0.19) | -1.66 | 0.096 | |
| | 20 | -0.30 (-0.83 to 0.02) | -0.59 (-1.18 to 0.17) | -0.38 | 0.705 | |
| | 30 | -0.23 (-2.05 to 0.31) | -1.15 (-2.74 to -0.19) | -0.76 | 0.450 | |
| | 40 | -1.04 (-3.03 to 0.66) | -1.47 (-4.17 to -0.63) | -0.68 | 0.496 | |
| | 50 | -2.07 (-4.92 to -1.16) | -3.60 (-6.21 to -1.19) | -0.30 | 0.762 | |
| | 60 | -4.22 (-7.00 to -2.09) | -6.59 (-10.78 to -3.46) | -0.60 | 0.545 | |
| | 70 | -6.45 (-9.69 to -3.73) | -9.53 (-13.74 to -5.72) | -0.45 | 0.650 | |
| | 80 | -7.84 (-14.28 to -4.87) | -11.89 (-18.94 to -5.94) | -0.76 | 0.450 | |
| | 90 | -9.79 (-16.41 to -5.72) | -13.01 (-23.79 to -7.76) | -0.68 | 0.496 | |
| | 100 | -10.77 (-15.25 to -4.82) | -14.51 (-24.89 to -10.14) | -0.98 | 0.326 | |
| Far Post-Test | 10 | -1.14 (-3.92 to -0.45) | -0.31 (-0.52 to 0.18) | -2.12 | 0.034 | |
| | 20 | -2.59 (-7.56 to -1.58) | -0.47 (-0.87 to -0.06) | -2.72 | 0.007 | |
| | 30 | -4.58 (-10.13 to -3.65) | -0.94 (-2.39 to 0.66) | -3.02 | 0.002 | * |
| | 40 | -7.43 (-10.19 to -5.35) | -2.22 (-3.36 to 0.90) | -3.02 | 0.002 | * |
| | 50 | -9.86 (-12.17 to -8.14) | -3.89 (-5.17 to -1.09) | -2.95 | 0.003 | * |
| | 60 | -12.34 (-14.67 to -11.34) | -6.71 (-8.09 to -3.05) | -3.25 | 0.001 | * |
| | 70 | -17.92 (-20.12 to -13.83) | -10.41 (-10.73 to -5.28) | -3.55 | 0.000 | * |
| | 80 | -24.62 (-26.01 to -20.13) | -13.07 (-14.60 to -11.87) | -3.17 | 0.001 | * |
| | 90 | -28.44 (-33.07 to -26.90) | -15.79 (-16.78 to -13.05) | -3.33 | 0.001 | * |
| | 100 | -30.25 (-33.41 to -27.59) | -16.48 (-19.44 to -12.21) | -3.48 | 0.001 | * |

The median of the relative angles of head rotation in the sagittal plane at each time point in the head-movement-focused group (Head) and the conventional training group (Conventional). Remaining significant results after Bonferroni correction are marked with an asterisk (*).

*$p < 0.005$ (Bonferroni corrected)

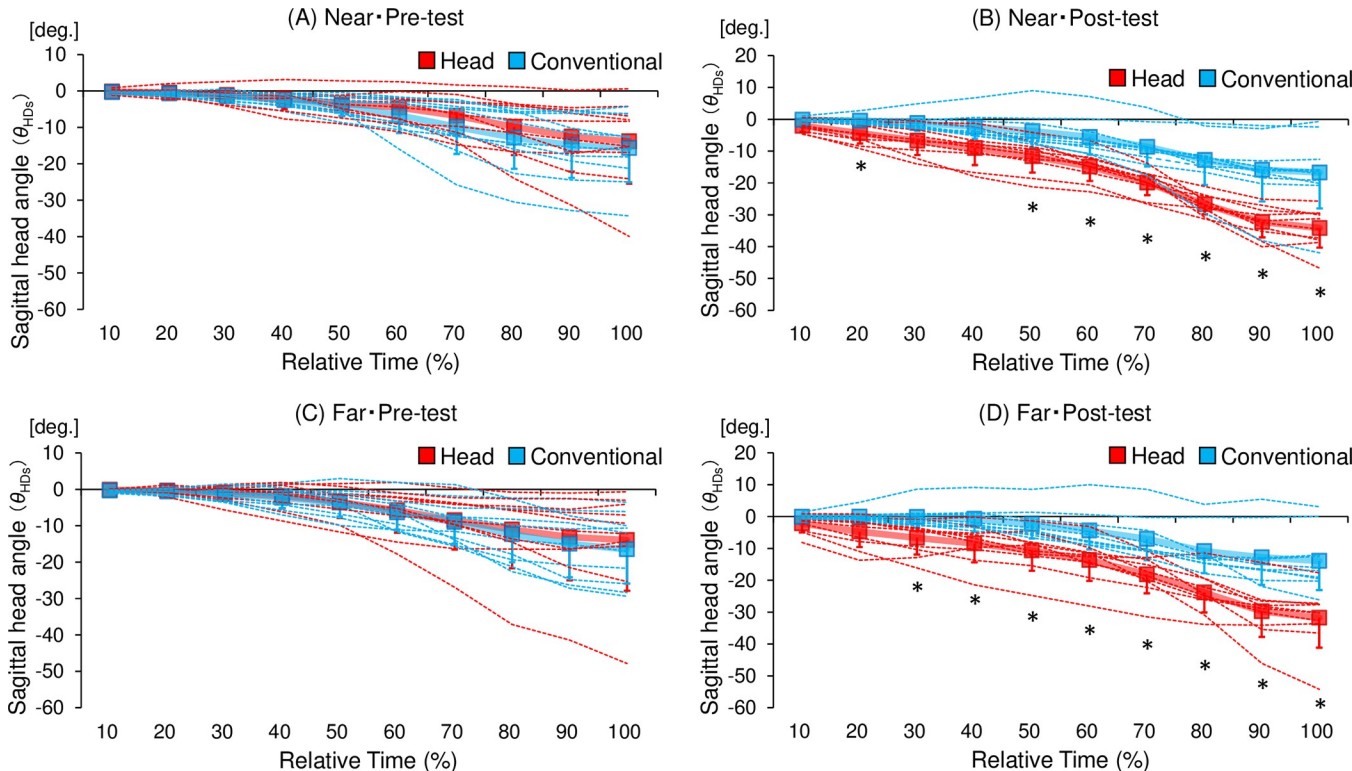

**Fig 5. Trajectories of the head rotation angles in the sagittal plane ($\theta_{HDs}$) relative to the time point at ball launching.** Comparison of the mean relative angles of head rotation in the sagittal plane at each time point between the head-movement-focused group (Head) and the conventional training group (Conventional). The red and blue colored bold lines represent the mean in each group. The red and blue colored thin dotted lines represent the values of the individuals in each group, and the bars represent the standard deviation. A negative value on the vertical axis in the figure indicates that the head was rotated more downward relative to the z-axis, while a positive value indicates that the head was rotated more upward. Remaining significant results after Bonferroni correction are marked with an asterisk (*). *$p < 0.005$ (Bonferroni corrected).

reaching and pointing to auditory targets [17]. Indeed, head rotation in the direction of the sound source during the sound presentation has been suggested to provide an accurate frame of reference for pointing control and contributes to the accuracy of pointing to the sound source [17]. In the present study, the head-movement-focused group turned their heads to a

**Table 4. Change in trunk ($\theta_{TRs}$) and head–trunk ($\theta_{HDs}-\theta_{TRs}$) angles in the sagittal plane.**

| | | | **Head** | | | **Conventional** | | |
|---|---|---|---|---|---|---|---|---|
| | | | **Pre-Test** | **Post-Test** | **Amount of Difference (deg.)** | **Pre-Test** | **Post-Test** | **Amount of Difference (deg.)** |
| Trunk ($\theta$TRs) peak-to-peak amplitude | Near | *Median (IQR)* | -7.70 (-8.88 to -6.28) | -11.23 (-12.31 to -9.30) | -3.45 (-4.65 to -1.79) | -7.57 (-10.33 to -6.77) | -8.79 (-10.00 to -7.10) | -0.23 (-1.30 to 0.70) |
| Trunk ($\theta$TRs) peak-to-peak amplitude | Far | *Median (IQR)* | -8.01 (-9.94 to -6.93) | -11.63 (-13.96 to -10.79) | -4.02 (-7.14 to -1.34) | -8.73 (-9.77 to -6.43) | -7.62 (-8.43 to -6.01) | 0.92 (-0.76 to 2.15) |
| Head–Trunk ($\theta$HDs–$\theta$TRs) peak-to-peak amplitude | Near | *Median (IQR)* | -5.75 (-13.68 to 0.03) | -22.43 (-30.96 to -17.39) | -16.07 (-21.54 to -14.24) | -7.26 (-12.68 to -5.46) | -9.87 (-12.64 to -8.84) | -1.44 (-7.24 to -0.57) |
| Head–Trunk peak-to-peak amplitude | Far | *Median (IQR)* | -4.98 (-11.43 to 0.32) | -20.29 (-28.08 to -12.15) | -13.82 (-18.69 to -9.63) | -9.10 (-17.39 to -4.60) | -10.56 (-13.38 to -8.31) | -3.04 (-4.54 to -0.98) |

The median of trunk and head–trunk angles in the sagittal plane in the pre-test, post-test, and the amount of difference between the head-movement-focused group (Head) and the conventional training group (Conventional).

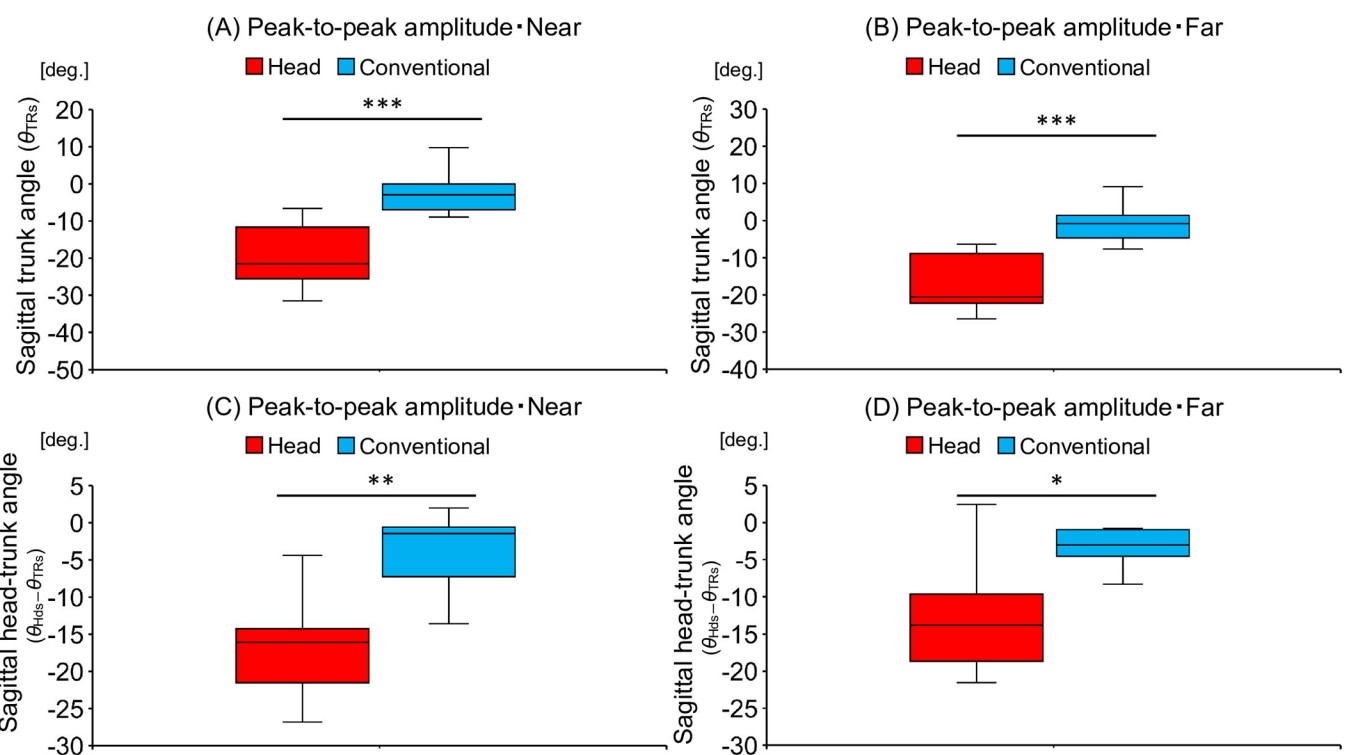

**Fig 6. The amount of difference in trunk rotation ($\theta_{TRs}$) and head–trunk ($\theta_{HDs}-\theta_{TRs}$) angles in the sagittal plane between pre- and post-tests.** Comparison of the amount of difference in the peak-to-peak amplitude of trunk rotation angle (A) in the near distance and (B) far distance between the head-movement-focused group (Head) and the conventional training group (Conventional). A negative value on the vertical axis in the figure indicates that the trunk was rotated more downward from the pre- to the post-test, while a positive value indicates that the trunk was rotated more upward. Comparison of the amount of difference in the peak-to-peak amplitude of head–trunk angle (C) in the near distance and (D) far distance between the head-movement-focused group (Head) and the conventional training group (Conventional). A negative value on the vertical axis in the figure indicates that the head was rotated more downward in relation to the trunk from the pre- to the post-test, while a positive value indicates that the head was rotated more upward in relation to the trunk. $^{*}p < 0.05$, $^{**}p < 0.01$, $^{***}p < 0.001$.

greater extent in the direction of the approaching ball than did the conventional training group, suggesting that they trapped the ball more accurately using a more accurate frame of reference. Furthermore, our results support the findings of a previous study [4], which revealed that blind football players use a greater downward head rotation when accurately trapping a ball. Considering these findings, our study suggests that the ball trapping skill is acquired by moving leftward and rightward to align the body along the ball's trajectory and following the ball with the face until the ball touches the feet.

Regarding the AE and VE, no significant changes were found in the difference between the head-movement-focused group and the conventional training group in the far distance. There are two possible reasons for this finding. First, in far distances, the participants had to rapidly identify the direction of the sound source based on the sound of the ball when it bounced on the ground so that they could move to the left or right and align with the ball's trajectory to trap it accurately. Previous studies have suggested that blind football players can rapidly and accurately localize the direction of horizontal sound sources, which can be improved by the long-term experience of playing blind football [1–3]. In the present study, the performance of the head-movement-focused group did not improve in the far distance because they had difficulty in rapidly localizing the sound direction during the 2-day skill acquisition training. Second, visual deprivation during the tasks may have influenced the variability of gait because the distance traveled was longer in the far distance than in the near distance. Previous studies have

reported that a more cautious walking strategy was adopted in the closed-eye condition than in the open-eye condition, showing slower walking speed, shorter stride length, and reduced balance in sighted individuals [33–37]. Based on these findings, the effects of the absence of vision on the stepping movement toward the left or right may have been greater in the far distance than in the near distance in the present study.

No significant difference was found between the groups regarding head angles in the horizontal plane; however, the head-movement-focused group showed larger downward head rotations in the sagittal plane than did the conventional training group. This suggests that the head-movement-focused group used a larger head rotation to localize the sound source more accurately than did the conventional-training group, which can be explained by two factors. First, head rotation provides auditory cues [38], which help humans improve sound localization accuracy [6–8]. Second, the head rotation allows listeners to localize a sound in front of their heads, where sound localization is known to be particularly accurate [32, 39, 40]. Collectively, our results explain how the head-movement-focused group localized the sound source (i.e., the ball) more accurately in front of their heads.

Recent findings by Valzolgher et al. [5, 41] demonstrated that implementing a multisensory-motor approach, involving reaching towards sound, is effective for training acoustic spatial perception. Training for sound localization in sighted individuals has shown that interaction with the sound source (reaching and head movements) is crucial for improving sound localization during spatial learning, suggesting a more active listening approach to better approximate the naturalistic experience with sounds [5]. These previous studies support our findings that implementing multisensory training, which involved interacting with the sound of the ball, improved sound localization and, consequently, ball-trapping skills in the head-movement-focused group.

In the pre-test phase, no significant difference was found between the groups regarding the relative angles of head rotation in the sagittal plane at each time point in the near or far distances. However, in the post-test phase, the relative angle of downward head rotations in the sagittal plane at each time point was significantly larger in the head-movement-focused group than in the conventional training group from the early time point after the ball launch to the ball-trapping time. These results suggest that the head-movement-focused group was encouraged to maintain the ball in a more consistent direction relative to the head throughout the skill acquisition phase. Previous studies have shown the head–ball coupling strategy during perceptual–motor tasks, where skilled baseball, cricket [18–21], and blind football players [4] couple the head direction with the ball movement to track the approaching target. Mann et al. reported that visuomotor tasks are controlled egocentrically, indicating the functional advantage of maintaining the target in a consistent direction relative to the head [21]. Additionally, the location of the sound source is mainly coded in the head-centered reference frame [22, 23]; thus, it is important to track the ball in an auditory-motor task, such as ball trapping, as is evident in blind football players [4]. The relative angle of head rotation in the head- movement-focused group observed in our study was similar to that observed in blind football players in a previous study [4]. These results indicate that the head-movement-focused group coupled downward head rotation with the movement of an approaching ball while keeping the ball in a consistent direction relative to the head.

The differences in the peak-to-peak amplitudes of the trunk and head–trunk angles in the sagittal plane were significantly larger in the head-movement-focused group than in the conventional training group. Additionally, downward head rotation may have resulted in a forward tilt of the trunk, leading to further downward head rotation. In a previous study [4], no difference in the angle of forward tilt of the trunk was found between skilled blind footballers and sighted individuals, implying that skilled players use a strategy of not tilting their trunk

forward to prepare for the following action, such as dribbling and passing after trapping the ball. However, in this study, we observed a greater forward tilt of the trunk and more downward head rotation in the head-movement-focused group than in the conventional training group. As the participants in this study were inexperienced beginners in playing blind football, the forward lean of the trunk observed in the head-movement-focused group could be used as a strategy to localize the sound closer to the head.

A recent study has indicated the efficacy of egocentric and allocentric training in view of multisensory-guided training for sound localization [42]. It is beyond the scope of this study to compare the effects of auditory-motor training on improving ball-trapping skills by setting up a group according to an allocentric reference frame with a group according to an egocentric reference frame. Further investigation is needed to investigate whether ball trapping skills can be improved through multisensory training using an allocentric reference frame, based on the auditory spatial bisection presented in previous findings [43–45].

A limitation of the present study is that the head-movement-focused group had a larger VE in the near distance compared to the conventional training group during the pre-test, resulting in insufficient standardization of the baseline skill level. This discrepancy may have contributed to the more effective skill acquisition observed in the head-movement-focused group. Further studies are needed to determine whether head-movement-focused training is effective for skilled participants. Additionally, previous studies have demonstrated the efficacy of auditory-motor training involving arm movement while wearing an audio device in the space around the body to improve the ability to localize sound sources [27, 45, 46]. The present study showed that learning to trap an approaching ball through head rotation decreases the distance (i.e., error) between the foot and ball. However, another limitation of our study is that we did not examine whether participants' ability to localize sounds improved by using a pointing method. Previous studies have used pointing at a sound source as an indicator to evaluate the accuracy of sound localization [17, 47]. Therefore, future research should calculate the angular error between the pointing and ball positions before and after skill acquisition to comprehensively assess the effect of head rotation on sound localization accuracy within an egocentric reference frame.

## Conclusion

The present study aimed to examine the effect of head rotation on ball-trapping skill acquisition in blind football through skill acquisition training over 2 days. Our study demonstrated a decreased in errors in ball trapping at near distances and with larger angles of downward head rotation in the head-movement-focused group compared to the conventional training group. These results suggest that larger head rotations towards the ball can localize the sound source more accurately than conventional ball-trapping methods. Our study also demonstrates the efficacy of head-movement-focused training in improving ball-trapping skills, where sound is localized by moving leftward and rightward to align the body with the ball's trajectory and follow the ball with the face. This method may help beginners in blind football maintain the ball in a consistent direction relative to the head while trapping it. Furthermore, this study discusses its impact on the field; head-movement-focused training could be helpful for blind players in other similar sports where auditory-motor skills are important, such as blind tennis and goalball.

## Supporting information

**S1 Table. Raw data of kinematic and performance variables in the pre- and post-tests, and the differences between pre- and post-tests.**
(XLSX)

**S2 Table. Raw data of relative angles of head rotations in the sagittal plane at each time point in the pre- and post-tests.**
(XLSX)

## Author Contributions

**Conceptualization:** Takumi Mieda, Masahiro Kokubu.

**Data curation:** Takumi Mieda, Masahiro Kokubu.

**Formal analysis:** Takumi Mieda, Masahiro Kokubu.

**Funding acquisition:** Takumi Mieda.

**Investigation:** Takumi Mieda.

**Methodology:** Takumi Mieda, Masahiro Kokubu.

**Project administration:** Masahiro Kokubu.

**Resources:** Masahiro Kokubu.

**Writing – original draft:** Takumi Mieda.

**Writing – review & editing:** Takumi Mieda, Masahiro Kokubu.

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
