## [Decision Letter · Decision Letter 0]

2 Jun 2024

PONE-D-24-11542Head movements affect skill acquisition for ball trapping in blind footballPLOS ONE

Dear Dr. Mieda,

Thank you for submitting your manuscript to PLOS ONE. After careful consideration, we feel that it has merit but does not fully meet PLOS ONE’s publication criteria as it currently stands. Therefore, we invite you to submit a revised version of the manuscript that addresses the points raised during the review process.

 **An external reviewer and myself reviewed your manuscript. Please carefully address the comments below and pay particular attention to some methodological points, some of which being also highly related to the motivation and hypotheses of the stud (e.g., definition of variables and relevance of these variables to the hypotheses). **

We look forward to receiving your revised manuscript.

Kind regards,

Dimitris Voudouris

Academic Editor

PLOS ONE

Journal Requirements:

JSPS KAKENHI (Grant Numbers: 22K17728, 19J13848) 

Cooperative Research Grant of Advanced Research Initiative for Human High Performance, University of Tsukuba

Additional Editor Comments:

At the moment there are several instances that require justificaiton and clarification. For instance, what exactly is meant by "performance" (e.g., line 179) and how is this calculated? I think these refer to the absolute and variable errors, but you should explain in detail how these were calculated. In addition, again regarding the AE and VE, it is unclear how these variables are related to the research question and what are the underlying hypotheses. This needs to be sharpened so that the conducted statistical tests can be justified. 

Another major point related to the same topic is the experimental manipulations of distance and speed. First, although the factor "distance" is included in the statistics, the factor "speed" is not. Why is "speed" a factor in the experiment but not in the statistics? Are there no hypotheses about this? Likewise, what are the hypotheses about the effect of "distance"? This needs to be explicitly stated in the Introduction. As far as I understand, the factor "direction" is not included in the design either. Please explain the methodological decision about the conditions, the underlying hypotheses and justify the statistical testing employed. 

A last major comment is about the comparisons across the timecourse. How is inflation of type I error being compensated here? I am also not confident that the description of the related analysis as well as the underlying hypotheses are explicit enough. Please revise accordingly. 

Line 101, Declaration of Helsinki: please mention which version you are referring to. If

Line 147, was the right foot also the dominant foot of all participants?

Line 161: if there are 12 trials during the pre-test, does it mean that there were 4 trials for each of the three speeds? Please clarify. Likewise, in line 177, why do you calcualte the average of all 12 trials and not separately per speed? See also another comment about the justification to use different speeds. 

Line 161: is each of the four blocks devoted to one of the four directions? If the directions were 'blocked', please explain why.

Line 180 and 182 (possibly also elsewhere): please specify what does a certain direction of effect mean. For instance, when the difference is 'smaller' (page 180), does it mean that the error was smaller in the pre- or in the post-test? Related to this, the figures show both negative and positive values for AE and VE. Please explain what does the sign of each value mean. The same comment applies to the other variables that have values of two signs (e.g., horizontal and saggital head angle, etc.). 

Line 196: here you describe an effect but the direction is unclear. I recommend specifying in the text which group had a larger VE. This applies to other related instances.

Lines 234ff: this paragraph can be written in a more concise way as there are several similar text passages that are being repeated. Perhaps you could combine the descirption about the absence of some effects in single sentences (e.g., head angle in horizontal plane in near and far conditions was not affected by group and this can be mentioned in a single sentence with both statistical results in brackets).  

Reviewers' comments:

Reviewer's Responses to Questions

**Comments to the Author**

1. Is the manuscript technically sound, and do the data support the conclusions?

Reviewer #1: Yes

2. Has the statistical analysis been performed appropriately and rigorously? 

Reviewer #1: Yes

3. Have the authors made all data underlying the findings in their manuscript fully available?

Reviewer #1: Yes

4. Is the manuscript presented in an intelligible fashion and written in standard English?

Reviewer #1: Yes

5. Review Comments to the Author

**Reviewer #1: **In general, the work seems well-written and detailed. I only have a few comments. However, I want to inform the authors that I am not an expert in the field.

The introduction is clearly written, however, in my opinion, it would be useful to distinguish between head movements used to orient towards a direction (e.g., towards the ball) and head movements that are not necessarily directed towards the ball but serve to inspect the acoustic space and increase the possibility of extracting acoustic cues (e.g., rotations to move the cone of confusion...). See for instance Valzolgher et al., (2022) in which the head-movements were used strategically to localize sounds sources.

The methods section lacks a discussion on how the sample size was determined. 10 vs 10 participants do not seem to be many; are there previous studies that justify this sample size?

Line 90: Did the participants play football? I understand they had no experience with blind football, but did they have experience with regular football?

Line 180: Smaller than what?

Line 182: More downward than what?

Figure 5: Why are there 6 lines per group? I understood there were 10 vs 10 participants. Was there some filtering?

Both in the introduction and discussion, the authors could consider adding references to the literature on motor training aimed at improving sound localization involving head movement. This could also help better describe the head strategies used in this study.

Valzolgher, C., Campus, C., Rabini, G., Gori, M., & Pavani, F. (2020). Updating spatial hearing abilities through multisensory and motor cues. Cognition, 204, 104409.

Valzolgher, C., Todeschini, M., Verdelet, G., Gatel, J., Salemme, R., Gaveau, V., ... & Pavani, F. (2022). Adapting to altered auditory cues: Generalization from manual reaching to head pointing. PLoS One, 17(4), e0263509.

Page 375: Regarding the egocentric-allocentric issue, this study might be of interest: Rabini, G., Altobelli, E., & Pavani, F. (2019). Interactions between egocentric and allocentric spatial coding of sounds revealed by a multisensory learning paradigm. Scientific reports, 9(1), 7892.

6. PLOS authors have the option to publish the peer review history of their article (what does this mean?). If published, this will include your full peer review and any attached files.

Reviewer #1: No

---

## [Author Response · Author response to Decision Letter 0]

9 Aug 2024

Dear Dr. Voudouris,

We wish to express our strong appreciation to the Editor for your constructive and insightful comments on our paper. 

The comments have helped us significantly improve the paper. We have carefully considered the editor’s comments regarding some methodological points, such as definition of variables and relevance of these variables to the hypotheses of this stud. We have incorporated changes that reflect the detailed suggestions you have generously provided.

We have now submitted the revised manuscript including the 'Response to Reviewers', 'Revised Manuscript with Track Changes' and 'Manuscript'.

We sincerely thank you for considering our paper for publication in PLOS ONE. We now hope that our paper will be suitable for publication in your journal.

Yours sincerely,

Takumi Mieda, Ph.D.

---

## [Decision Letter · Decision Letter 1]

17 Sep 2024

PONE-D-24-11542R1Head movements affect skill acquisition for ball trapping in blind footballPLOS ONE

Dear Dr.  Mieda,

Thank you for submitting your manuscript to PLOS ONE. After careful consideration, we feel that it has merit but does not fully meet PLOS ONE’s publication criteria as it currently stands. Therefore, we invite you to submit a revised version of the manuscript that addresses the points raised during the review process by reviewer #2.

We look forward to receiving your revised manuscript.

Kind regards,

Riccardo Di Giminiani

Academic Editor

PLOS ONE

Journal Requirements:

Reviewers' comments:

Reviewer's Responses to Questions

**Comments to the Author**

1. If the authors have adequately addressed your comments raised in a previous round of review and you feel that this manuscript is now acceptable for publication, you may indicate that here to bypass the “Comments to the Author” section, enter your conflict of interest statement in the “Confidential to Editor” section, and submit your "Accept" recommendation.

Reviewer #1: (No Response)

Reviewer #2: (No Response)

2. Is the manuscript technically sound, and do the data support the conclusions?

Reviewer #1: Yes

Reviewer #2: Yes

3. Has the statistical analysis been performed appropriately and rigorously? 

Reviewer #1: Yes

Reviewer #2: I Don't Know

4. Have the authors made all data underlying the findings in their manuscript fully available?

Reviewer #1: Yes

Reviewer #2: Yes

5. Is the manuscript presented in an intelligible fashion and written in standard English?

Reviewer #1: Yes

Reviewer #2: Yes

6. Review Comments to the Author

Reviewer #1: I have read the revised version of the manuscript and I am satisfied with the changes made by the authors, which bring more clarity to the manuscript.

Reviewer #2: General Comments

Congratulations, the topic of the paper is interesting. The manuscript is well written, the content is clearly presented, as are the results, figures and tables.

Considering the corrections and changes made by the authors, following the comments of the other reviewers, this study has improved the construction of the hypothesis, and results' clarity, understanding.

The Authors investigated the incidence of head movements during the acquisition of the ball trapping task in blind football. The study provides important insights that could help the training methods of bling football, especially during the learning of ball location in space by beginners.

The specific hypothesis is well formulated, and the rationale is consistent with the current literature.

However, not being an expert in the field, I would like to ask you for some clarification on the methods and I have some minor comments that, I think, can be addressed to improve the understanding of the article.

Minor comments:

- In the procedure section, what was the approximate duration of each day of skill acquisition training, did the duration vary for the two groups and between participants within each group? If there were differences, please specify in the manuscript text

- Also in the procedure section: how much time elapsed between skill acquisition and pre-test on day 1, and between post-test and acquisition on day 2? In addition, if possible I would ask to specify whether the two days, which the procedure refers to, were consecutive or non-consecutive.

- In the apparatus section, when describing the experimental set-up, I suggest specifying in the manuscript the distance between the apparatus for throwing the ball and the participants (although shown in figure.1) Since it represents an important data in the structure of the protocol, as the distance could influence the final speed of the ball and the orientation of the participants' head.

- From the 158 to the 161, the period is not apparent to me, it seems almost confusing. Defined that speed is not a dependent variable of your study. I cannot understand if it is indicated that a randomization of the velocities was made to prevent the learning effect of the subjects, or that the velocities were not randomized because it was not a variable of study interest. In both cases, the authors could clarify the sentence.

- Were the combinations of speed and trajectory of the ball thrown administered in a randomized order, with a pre-defined order for all participants, or was it chosen independently by the experimenter who threw the ball? Was the experimenter who moved the device always the same in all tests? Please clarify and specify these aspects if possible.

- At line 193 it is not clear whether the familiarization tests were carried out at a fixed speed and direction, or also for the 12 familiarization tests they represent the combination of heights and speeds? If possible, better specify the concept

-Line 194 when reporting of the 48 tests (12 tests 4 blocks) during the acquisition of the ball catching ability, the authors mean that the 12 combinations (3 directions x 4 heights) were repeated 4 times. If so, were the 4 blocks all administered in the same order or were they randomly ordered? Please clarify the concept.

7. PLOS authors have the option to publish the peer review history of their article (what does this mean?). If published, this will include your full peer review and any attached files.

Reviewer #1: No

Reviewer #2: **Yes: **Stefano La Greca

---

## [Author Response · Author response to Decision Letter 1]

11 Oct 2024

Dear Dr. Giminiani,

We greatly appreciate the time and effort you and the reviewers have dedicated to providing detailed and constructive feedback to strengthen our paper.

We have incorporated changes that reflect the detailed suggestions you have generously provided. 

We also hope that our edits and responses will satisfactorily address all the issues, queries, and concerns you and the reviewer have noted.

We have now submitted the revised manuscript including the 'Response to Reviewers',

'Revised Manuscript with Track Changes' and 'Manuscript'.

We sincerely thank you for considering our paper for publication in PLOS ONE. 

We hope that it now be deemed suitable for publication in your journal and look forward to hearing from you at your earliest convenience.

Yours sincerely,

Takumi Mieda, Ph.D.

---

## [Decision Letter · Decision Letter 2]

23 Oct 2024

Head movements affect skill acquisition for ball trapping in blind football

PONE-D-24-11542R2

Dear Dr. Mieda,

We’re pleased to inform you that your manuscript has been judged scientifically suitable for publication and will be formally accepted for publication once it meets all outstanding technical requirements.

Kind regards,

Riccardo Di Giminiani

Academic Editor

PLOS ONE

Additional Editor Comments (optional):

Reviewers' comments:

Reviewer's Responses to Questions

**Comments to the Author**

1. If the authors have adequately addressed your comments raised in a previous round of review and you feel that this manuscript is now acceptable for publication, you may indicate that here to bypass the “Comments to the Author” section, enter your conflict of interest statement in the “Confidential to Editor” section, and submit your "Accept" recommendation.

Reviewer #2: All comments have been addressed

2. Is the manuscript technically sound, and do the data support the conclusions?

Reviewer #2: (No Response)

3. Has the statistical analysis been performed appropriately and rigorously? 

Reviewer #2: (No Response)

4. Have the authors made all data underlying the findings in their manuscript fully available?

Reviewer #2: (No Response)

5. Is the manuscript presented in an intelligible fashion and written in standard English?

Reviewer #2: (No Response)

6. Review Comments to the Author

Reviewer #2: Dear authors,

I am satisfied with your responses. The article has improved, and so has the clarity of the methods. I think the article is ready to be published.

7. PLOS authors have the option to publish the peer review history of their article (what does this mean?). If published, this will include your full peer review and any attached files.

Reviewer #2: **Yes: **Stefano La Greca

---

## [Editor Report · Acceptance letter]

28 Oct 2024

PONE-D-24-11542R2 

PLOS ONE

Dear Dr. Mieda, 

I'm pleased to inform you that your manuscript has been deemed suitable for publication in PLOS ONE. Congratulations! Your manuscript is now being handed over to our production team.

Kind regards, 

on behalf of

Prof Riccardo Di Giminiani 

Academic Editor

PLOS ONE